# The Human Nature of Generative AIs and the Technological Nature of Humanity: Implications for Education

**Jon Dron** 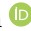

Faculty of Science and Technology, Athabasca University, Athabasca, AB T9S 3A3, Canada; jond@athabascau.ca

**Abstract:** This paper analyzes the ways that the widespread use of generative AIs (GAIs) in education and, more broadly, in contributing to and reflecting the collective intelligence of our species, can and will change us. Methodologically, the paper applies a theoretical model and grounded argument to present a case that GAIs are different in kind from all previous technologies. The model extends Brian Arthur's insights into the nature of technologies as the orchestration of phenomena to our use by explaining the nature of humans' participation in their enactment, whether as part of the orchestration (hard technique, where our roles must be performed correctly) or as orchestrators of phenomena (soft technique, performed creatively or idiosyncratically). Education may be seen as a technological process for developing these soft and hard techniques in humans to participate in the technologies, and thus the collective intelligence, of our cultures. Unlike all earlier technologies, by embodying that collective intelligence themselves, GAIs can closely emulate and implement not only the hard technique but also the soft that, until now, was humanity's sole domain; the very things that technologies enabled us to do can now be done by the technologies themselves. Because they replace things that learners have to do in order to learn and that teachers must do in order to teach, the consequences for what, how, and even whether learning occurs are profound. The paper explores some of these consequences and concludes with theoretically informed approaches that may help us to avert some dangers while benefiting from the strengths of generative AIs. Its distinctive contributions include a novel means of understanding the distinctive differences between GAIs and all other technologies, a characterization of the nature of generative AIs as collectives (forms of collective intelligence), reasons to avoid the use of GAIs to replace teachers, and a theoretically grounded framework to guide adoption of generative AIs in education.

**Keywords:** AI; education; technology



## 1. Introduction

The rapid growth in power and consequent use of generative AIs (GAIs) in recent years, especially since the release of ChatGPT in 2022, has raised or brought to prominence a wide range of concerns among educators, from student uses of GAIs for cheating [1] to teaching job losses and transformations [2] to fears about GAIs' effects on learners' sensemaking and socialization [3,4]. Equally, many have seen great promise in the use of such tools to support, engender, or reduce costs of learning [1,3,5,6]. However, there has been little that situates the discussion in theory, and still less that addresses both the educational and the technological underpinnings of the phenomenon. Most if not all commentators have treated GAIs as simply species of technologies that follow the same patterns and behaviours of other technologies and/or their roles in socio-technical systems, treating them as tools that we might use like any other. This paper challenges such a view. It presents a novel, theoretically grounded argument that GAIs represent an entirely new phenomenon in the history of our relationship with technologies, centering around the key observation that, for the first time, the technologies we have created are capable of something that closely resembles the soft, original, idiosyncratic, creative technique that

was formerly the sole domain of human beings. We can no longer lay exclusive claim to the creative use of technologies.

This paper examines the consequences of this phenomenon as they relate to what we learn, how we learn, and, ultimately, the nature of human cognition itself. Concerns are expressed that, if we habitually and at scale offload not just the teaching and learning tasks that humans perform but the processes of sensemaking and creative application that underpin the doing of them to something that is not human, there are risks of losing much of the relational, tacit, and socializing value of education, of diluting the cultural roles played by educational systems, and of diminishing the cognitive capabilities of future generations because our descendants may not develop the soft skills that GAIs replace.

The paper presents and critically examines a number of approaches that may reduce the harm while leveraging the benefits of GAIs. It begins by briefly summarizing the theoretical basis for its arguments before moving on to its implications as they relate to generative AIs, concluding with a discussion of ways to limit their potentially harmful consequences.

## 2. Methodology

This paper presents a deductive and inferential grounded argument, synthesizing the literature from a number of fields, including that on complexity theory, the philosophy of technology and socio-technical systems, neuroscience, educational theory, and machine learning, to present the case for a new and productive way of understanding GAIs and their roles in learning. It is not a systematic review. It applies a theory of technology drawn from the author's book, *How Education Works: Teaching, Technology, and Technique* [7], that extends Brian Arthur's understanding of technologies as assemblies of orchestrations of phenomena to our use [8] to focus on the roles we play in their enactment, individually and collectively. It provides not only a means to describe the educational process but an explanation of its nature and its products in technological terms. The theory is situated in a broadly complexivist [9] tradition of educational research, related to Fawns's view of education and technology as entangled systems [10], connectivist models of learning [11,12], and distributed cognition [13], amongst others. Given that among the central premises of such theories it is stated that learning is highly situated, complex, and unpredictable at a detailed level, and that the ways technologies may develop are inherently unprestatable [14], it is a limitation of the argument that any specific predictions it makes, beyond those in the immediate future, may and most likely will be wrong. Although some possible future consequences will be presented, the intent of the argument is thus not to predict the future but to provide a way of understanding that future as it unfolds.

## 3. Theoretical Model

This section is a summary of the relevant aspects of the theory presented in *How Education Works* [7]. Any unreferenced claims in this section should be assumed to derive from the book itself.

We are a part of our technologies and they are a part of us. They are not just tools we use but intrinsic parts of our cognition and ways of being [15]. Equally, we are not just users of them but parts of their assembly, inasmuch as the techniques that we use when operating them are as much technologies as computers and books. Whether we are sole orchestrators (for instance, in the use of language, singing, or mental arithmetic) or parts of a broader orchestration (for instance, the ways in which we operate power stations, enact regulations, or simply turn on a light), we are, through technique, active participants in their orchestration. Sometimes, for instance, when spelling a word or telling the time, we are mechanical parts of their orchestration who must play our roles correctly. I describe these fixed techniques as *hard*, in the sense of being, when enacted correctly, inflexible and invariant. Sometimes, such as when writing a sentence or designing software, we are the orchestrators, using an idiosyncratic technique to create new technologies such as academic papers, stories, and apps. I describe such techniques as *soft*, in the sense of being flexible and variable. Most of the time, we are both orchestrators and the orchestrated, using a

mix of hard and soft technique, because almost all technologies are assemblies of other technologies [8], some of which invite our own orchestration, and some of which demand that we participate correctly in theirs. For example, to play a musical instrument we must train ourselves to place our fingers, breathe, shape our lips, and tune the instrument correctly, but the things we usually value most highly are the idiosyncratic ways we play the notes.

Each new technology (including soft technique) creates adjacent possible empty niches into which further technologies may step, and relies on those created by its predecessors [16]. New technologies are not just derived from but must fit in with others that already exist; we virtually never see the wholesale replacement of one type of technology with another, in part because most technologies use services of others (cars need roads, pens need paper, etc.) and/or are made from them [8], and in part because of the natural dynamics of pace layering [17]: that in all systems, be they natural or artificial, the larger and slower moving tend to influence the smaller and faster moving more than vice versa [18]. While some small things, en masse, may be highly disruptive (viruses, say, or locusts), this is because the small parts are members of a larger collective that can be treated at a system level as a single entity. Pace layering is a facet of a larger family of path dependencies, where what has occurred in the past both enables and constrains what may occur in the future. The large and slow-moving nearly always exist prior to any individual smaller or faster phenomenon precisely because they are slower to change; they provide the background to which smaller, faster changing parts must develop and adapt. This may in turn be reframed in technological terms: technologies that are harder and more invariant, by definition, change more slowly than softer, more flexible, and malleable technologies, which must fit in and around their constraints. Thus, the harder a technology, the more embedded in relation to others it may be, and the greater its influence in a technological system.

None of us could be smart alone and no one learns alone. It is almost entirely through our technologies, from language to doorknobs, that we are able to participate in one another's cognition and, ultimately, in the ever expanding intertwingled collective intelligence of our species. We stand not only on the shoulders of giants but on those of all who came before, and of all the people we have directly or indirectly (through their creations) encountered. As we participate in our cultures, being a part of, creating, building, adapting, and assembling technologies, we all contribute to the learning of others—for better or worse—and so we and our technologies co-evolve in an endless iterative and recursive cycle leading (globally, though not always locally) to greater complexity, greater diversity, and greater technological capability [19]. We and our cognition exist for and by means of our societies that exist for and by means of us, mediated through the technologies we create and enact.

Although such learning is embedded and is simply inevitable as a result of living with other people, thanks to the complexity and diverse needs of modern societies we often need to formalize the learning of techniques (soft and hard) through a set of technological processes we normally describe as *education*. Education is not just a set of pedagogical techniques performed by those we label as teachers, but by all the participants involved: by authors of textbooks, designers of classrooms, creators of test banks, manufacturers of whiteboards, members of academic boards, creators of regulations, other students, and, above all, by learners themselves. Institutions, including their processes, regulations, structures, and infrastructures, as well as the many kinds of interactions between the people in them, teach at least as much as those who are formally designated as teachers.

Education plays a role in the development of values and attitudes that go far beyond the technological but, at its heart, and fundamental to supporting these other roles, it is concerned with building the cognitive gadgets [20] needed to participate in the many technologies of our societies. This includes training for hard skills (spelling, performing lab experiments correctly, following citation standards, using rules of logic in causal reasoning, etc.) as well as the development of soft skills (composition, problem solving, rhetoric, research design, musical expression, etc.). In other words, the process of education is largely

concerned with creating, fostering, and developing technique, including technologically mediated knowledge. Every subject includes hard and soft elements of technique in varying measures: harder subjects, as the name implies, tend to focus more on hard technique, and softer subjects focus more on soft technique. Through education we learn the technologies of our many overlapping cultures, from methods of scientific experimentation to rituals of religion or the mechanics of political systems. Often, we need to develop literacies, which may be thought of as the prerequisite cognitive gadgets that we need in order to participate in other technologies of our cultures. Education itself is fundamentally technological in character, involving the assembly of methods, principles, processes, physical tools, cognitive tools, buildings, networks, and countless other technologies to enact a process in which there are countless co-participants. No one ever teaches alone, and no one ever learns alone. From obvious teachers like the authors of textbooks or directors of videos to significant players like architects or furniture designers, and above all the learners themselves, at least thousands of people participate directly or indirectly in any formal teaching process.

Technologies help us to solve problems or to create opportunities, but we are the solvers and the opportunity takers. As the technologies of our societies evolve, so too do the needs for the skills to use them and thus, so too do they become incorporated into what is taught in our institutions, in an endless cycle of renewal. Until very recently, though they nearly always support and enable the development of soft technique, the physical and virtual technologies we have created have only ever been hard, leaving the softer ways of assembling, using, and creating them to humans. Indeed, we could barely call them technologies at all were it not that there was something consistent and invariant about them. It is only our own orchestration of them that could, till now, rightly be described as soft. The development of generative artificial intelligence has changed that.

## 4. The Distinctive Nature of Generative AIs

GAIs, notably but not exclusively in the form of large language models (LLMs), have now developed to a point that their output closely resembles and often exceeds what humans could do unaided, performing tasks that appear to be the result of soft cognitive processes much like our own. In fact, this is because that is, to a large extent, almost exactly what they are. The "intelligence" of LLMs is almost entirely composed of the reified soft creations of the (sometimes) hundreds of millions of humans whose data made up their training sets albeit that it is averaged out, mashed up, and remixed. LLMs are essentially a technological means of mining and connecting the collective intelligence [21] of our species.

For more than a decade, conversational agents have been available that, within a constrained context, have regularly fooled students that they are human, albeit making sometimes embarrassing or harmful mistakes due to their hitherto relatively limited training sets [22] and seldom fooling the students for very long. The main thing that has changed within the past few years is not so much due to the underlying algorithms or machinery, though there have been substantial advances (such as transformers and GPU improvements), but to the exponentially increasing size of the language models. The larger the training set, the greater the number of layers and vectors, and the larger the number of parameters, the more probable that the model will not only be able to answer questions but do so accurately and in a human-like way. Their parameters (directly related to the number of vectors and layers) provide an approximate measure of this. Open AI's GPT-3, released in 2022, has around 175 billion parameters, while Google's slightly earlier BERT has "only" 340 million. However, both are dwarfed by GPT-4, released in 2023, which is estimated to use closer to 100 trillion parameters, being trained on a data set representing a non-trivial proportion of all recorded human knowledge [23]. It is because of this that modern LLMs appear to be capable of mimicking and, in many cases, that the quality of their outputs exceed all but the highest achievements in human cognition including inference [24] and creativity [25,26].

Some (e.g., [27,28]) have even tried to make the case that a GAI such as ChatGPT-4 is now at least close to being an AGI (artificial general intelligence), using measures of human intelligence and creativity as evidence. I disagree, for reasons that will matter in the discussion that follows. These measures were chosen by researchers to determine the extent to which a *human* is intelligent or creative; they rely on indicators that usually correlate with what we normally recognize as intelligent, creative behaviour in a human being. In so doing they assume, as a baseline, that the agents they are testing *are* both creative and intelligent, so the tests are a means to compare one human with another on a scale, and are not absolute standards and certainly not a proxy for the cognitive skills themselves.

To measure something requires there to be attributes that we can define precisely enough to measure. Unfortunately, both intelligence and creativity are extremely fuzzy culturally embedded concepts with meanings that shift according to context and that drift over time [29]. We know them when we see them but, if called upon to define them, we invariably come up with definitions that are too narrow or too broad, and that admit exceptions or that include things we would not see as anything similar to our own. This is inevitable because intelligence and creativity are identified by family resemblances [30], not a fixed set of defining characteristics. We see in others signals of aspects we see in ourselves, recognizing shared physical and behavioural characteristics, and then extrapolate from these observations that they emerge from the same kind of entity. The signals are, however, not the signified. The meanings we give to "intelligence" or "creativity" are social constructions representing dynamic and contextually shifting values, not fixed natural phenomenon like the boiling point of water or gravity. In them we find reflections of our own ever-evolving and socially constructed identities, not laws of nature. While we can make general inferences from correlational data, they cannot reliably predict behaviour in any single instance [31]. Tests of intelligence or creativity are broadly predictive of what we recognize as intelligent or creative behaviour, but they are highly susceptible to wide fluctuations at different times that depend on many factors such as motivation, emotion, and situation [32].

Just because the output of an LLM closely resembles that of a human does not mean it results from the same underlying mechanisms. For instance, some of an LLM's apparent creative ability is inherent in the algorithms and data sets it uses; LLMs have vastly greater amounts of reified knowledge to draw from than any individual human, and the fact that they can operate at all depends on their capacity to connect and freely associate information from virtually any digital source, including examples of creativity. If this is how we choose to define creativity then, of course, they can be very creative. It is, though, inappropriate to directly compare the intelligence, wisdom, or creativity of AIs and humans, at least in their current forms, because, even if some of the underlying neural nets are analogous to our own, they are not like us, in ways that matter when they are a part of the fabric of our own cognitive, social, and emotional development.

Unlike humans, the current generation of LLMs have not learned about the world through interactions with it, as independent and purposeful agents interacting with other independent and purposeful agents. Their pasts are invented for them, by us, and their purposes are our purposes, not their own. Although we might metaphorically describe their behaviours as goal-seeking, this is because that is how they are programmed, not because they possess goals themselves. LLMs have no intentions, nothing resembling consciousness, no agency, and no life history. They have no meaningful relationships with us, with one another, or with the tokens they unknowingly assemble into vectors. Though there may be much sophistication in the algorithms surrounding them, and impenetrable complexity in the neural networks that drive them, at their heart they just churn out whatever token (a word, a phrase, musical notes, etc.) is most likely to occur next (or, in some systems, whatever comes previously, or both), given the prompt they are given.

Perhaps something similar is true of human beings; we certainly make decisions before we are conscious of having done so and many if not all of our intentions are pre-conscious [33]. Also, like us, LLMs are prediction machines [34] and they do appear to

make such predictions in a similar manner. However, as Clark [35] argues, it is not possible to jump from this to a full explanation of human thought and reason, let alone intentional behaviour. Even if there are closer similarities with our own minds, the stuff that such minds deal with is fundamentally different. Most significantly and unsurprisingly, because all it has learned has been the processed signals humans (mostly intentionally) leave in the digital world, an LLM is nothing *but* signals, with nothing that is signified underneath. The symbols have no meaning, and there is no self to which they could relate. Current systems have no concept of whether the words or media they churn out make sense in the context of the world, only whether they are likely to occur in the context of one another. If part of their output is a hallucination, then all of it is. The machines have no knowledge, no concepts, and no sense of how anything works in the context of a self because there is no identity, no purposive agent, and no being in the world to which the concept could relate. This may change as embodied AIs become more common and sophisticated but, even then, unless perhaps they are brought up like humans in a human society (a possibility fraught with huge ethical and practical concerns), they will be utterly unlike us.

Some might argue that none of this is important. If it walks like a duck, squawks like a duck, and flies like a duck then, to all intents and purposes, we might as well call it a duck. This is, again, to mistake the signal for the signified. While the output of an LLM may fool us into thinking that it is the work of an actual human, the creative choices we most value are expressions of our identity, our purposes, our passions, and our relationships to other people. They are things that have meaning in a social context, and are things that are situated in our lives and the lives of others. It matters so much, for example, that a piece of work was physically written by Gustav Mahler that someone was willing to pay over USD 5m for the handwritten score of his Second Symphony. We even care about everyday objects that were handled by particular humans; an inexpensive mass-produced guitar used by John Lennon in some of his early songwriting, for instance, can sell for roughly USD 2.4m more than one that was not. From a much loved piece of hand-me-down furniture to the preservation of authorship on freely shared Creative Commons papers, our technologies' value lies as much as or more than in their relationship to us, and how they mediate relationships between us, as in their more obvious utilitarian functions. More prosaically, we are normally unwilling to accept coursework written by an AI when it is presented as that of a student, even though it may be excellent, because the whole point is that it should have contributed to and display the results of a human learning process. This is generalizable to all technologies; their form is only meaningful in relationship to other things, and when humans participate in the intertwingled web that connects them. It is not just our ability to generate many ideas but our ability to select ones that matter, to make use of them in a social context, to express something personal, and to share something of ourselves that forms an inextricable part of their value. The functional roles of our technologies, from painting techniques to nuts and bolts to public transit systems, are not ends in themselves; they are meant to support us in our personal and social lives.

Despite appearances, we are thus little closer to an AGI now than we were 10 years ago. In fact, as Goertzel [36] observed back then, we still struggle to define what "intelligence" even means. The illusion of human-like intelligence, though, being driven by the reified collective knowledge of so many humans and, for most large models, being trained and fine-tuned by tens or hundreds of thousands more, is uncanny. To a greater extent than any previous technology, LLMs black-box the orchestration of words, images, audio, or moving images, resulting in something remarkably similar to the soft technique that was hitherto unique to humans and perhaps a few other species. Using nothing but those media and none of the thinking, passion, or personal history that went into making them, they can thus play many soft, creative, problem-solving, generative roles that were formerly the sole domain of people and, in many cases, substitute effectively for them. More than just tools, we may see them as partners, or as tireless and extremely knowledgeable (if somewhat unreliable) coworkers who do so for far less than the minimum wage. Nowhere is this more true, and nowhere is it more a matter of concern, than in the field of education.

## 5. GAIs and Education

The broader field of AI has a long history of use in education for good reason. Education is a highly resource-intensive activity demanding much of its teachers. We have long known that personal tuition offers a two-sigma advantage when compared with traditional classroom methods [37] but, for most societies, it is economically and practically impossible to provide anything close to that for most students. There is therefore great appeal to automating some or all of the process, either to provide such tuition or to free up the time of human teachers to more easily do so. The use of automated teaching machines stretches back at least 70 years [38,39], though it would be difficult to claim that such devices had more than the most rudimentary intelligence. AIs now support many arduous teaching roles. For instance, since at least as long ago as the 1990s, auto-marking systems using statistical approaches to identify similarity to model texts [40], or latent semantic analysis with examples trained using human-graded student work [41], have been able to grade free-text essays and assignments at least as reliably and consistently as expert teachers. For at least 20 years, some have even been able to provide formative feedback, albeit normally of a potted variety selected from a set of options [42]. Use of intelligent tutoring systems that adapt to learner needs and that can play some (though never all) roles of teachers, such as selecting text, prompting thought or discussion, or correcting errors, goes back even farther, including uses of expert systems [43], adaptive hypermedia that varies content or presentation or both according to rules adapted to user models [44], as well as rule-based conversational agents (that might now be described as bots) mimicking some aspects of human intelligence from as far back as the 1960s, such as Coursewriter [45], ELIZA [46], or ALICE [47,48]. Discriminative AIs performing human-like roles of classification have seen widespread use in, for example, analyzing sentiment in a classroom [49], identifying engagement in online learning [50], and identifying social presence in online classes [51]. From the algorithms of search engines such as Google or Bing to grammar-checking, autocorrect, speech-to-text, and translation tools, the use of AIs of one form or another for performance support and task completion has been widespread for at least 25 years, and nowhere more than in education.

For all of the sometimes mixed benefits AIs have brought, and for all of the ways they have benefited students and teachers, until now they have been tools and resources that are parts of our own orchestrations, not orchestrators in their own right. They had neither the breadth of knowledge nor the range of insight needed to respond to novel situations, to act creatively, or to fool anyone for long that they are human. Now that this is possible, it has opened up countless new adjacent possibilities. There has been an explosion of uses and proposed uses of GAIs in education, both by students and by teachers, performing all these past roles and more [5,52]. For teachers, GAIs can augment and replace their roles as Socratic tutors, providers of meaningful feedback, participants in discussions, and curriculum guides [53,54]. For students they can write assignments, perform research, summarize documents, and correct improper use of language [55]. These examples merely scratch the surface of current uses.

The effects of GAIs on our educational systems have already been profound. At the time of writing, less than a year after the meteorically successful launch of ChatGPT, recent surveys suggest that between 30% (https://www.intelligent.com/nearly-1-in-3-college-students-have-used-chatgpt-on-written-assignments/ accessed on 25 November 2023) and 90% (https://universitybusiness.com/chatgpt-survey-says-students-love-it-educators-not-fans/ accessed on 25 November 2023) of students are using it or its close cousins to assist with or often write their assessed work. Teachers, though mostly slower to jump on the bandwagon, are using these tools for everything from the development of learning outcomes and lesson plans to intelligent tutors who interact with their students, and they are scrambling to devise ways of integrating GAIs with curricula and the course process [52]. Already, in some cases it may therefore be the case that the bulk of both the students' and the teachers' work is done by a GAI. This has a number of significant implications.

Teachers, be they human or AI, are not only teaching the pattern of the cloth; they are teaching how to be the loom that makes it or, as Paul [56] puts it, the mill as well as the grist of thought. Although the language of education is typically framed in terms of learning objectives (what teachers wish to teach) and learning outcomes (what it is hoped that students will learn), there is always far more learning that occurs than this; at the very least, whether positive or negative, students learn attitudes and values, approaches to problem solving, ways of thinking, ways of relating to others in this context, motivation, and ways of understanding. It is telling, for instance, that perceived boredom in a teacher results in greater actual boredom in students [57]. Similarly, approaches to teaching and structural features of educational systems that disempower learners create attitudes of acquiescence and detract from their intrinsic motivation to learn [58–60]. Equally, the enthusiasm of a teacher plays an important role in improving both measured learning outcomes and attitudes of students towards a subject [61,62]. Such attitudinal effects only scratch the surface of the many different kinds of learning, ways of connecting ideas, and ways of being that accompany any intentional learning that involves other people, whether they are designated teachers, authors of texts, or designers of campuses. Often, teachers intentionally teach things that they did not set out to teach [63]. There are aspects of social and conceptual relationships and values that matter [59], idiosyncratic ways of organizing and classifying information, ethical values expressed in actions, and much, much more [64]. There is a hidden curriculum underlying all educational systems [65] that, in part, those educational systems themselves set out to teach, that in part is learned from observation and mimicry, and that in part comes from interacting with other students and all of the many teachers, from classroom designers to textbook authors, who contribute to the process, as well as all the many emergent phenomena arising from ways that they interact and entwine. Beyond that, there is also a tacit curriculum [66] that is not just hidden but that cannot directly be expressed, codified, or measured, which emerges only through interaction and engagement with tasks and other people.

The tacit, implicit, and hidden curricula are not just side-effects of education but are a part of its central purpose. Educational systems prepare students to participate in the technologies of their various cultures in ways that are personally and socially valuable; they are there to support the personal and social growth of learners, and they teach us how to work and play with other humans. They are, ultimately, intended to create rich, happy, safe, caring, productive societies. If the means of doing so are delegated to simulated humans with no identity, no history, no intention, no personal relationship, and with literally no skin in the game, where a different persona can be conjured up through a single prompt and discarded as easily, and where the input is an averaged amalgam of the explicit written words (or other media) of billions of humans, then students are being taught ways of being human by machines that, though resembling humans, are emphatically *not* human. While there are many possible benefits to the use of AIs to support some of the process, especially in the development of hard technique, the long-term consequences of doing so raise some concerns.

*The End and the Ends of Education*

We are at the dawn of an AI revolution to which we bring what and how we have learned in the past, and so—like all successful new technologies—we see great promise in the parts of us and the parts of our systems they can replace. All technologies are, however, Faustian bargains [67] that cause as well as solve problems, and the dynamics of technological evolution mean that some of those problems only emerge at scale when technologies are in widespread use. Think, for example, of the large-scale effects of the widespread use of automobiles on the environment, health, safety, and well-being.

Generative AIs do not replace entire educational systems; they fit into those that already exist, replacing or augmenting some parts but leaving others—usually the harder, larger-scale, slower-changing parts, such as systems of accreditation, embedded power imbalances, well-established curricula, and so on—fully intact, at least for now. They are

able to do so because they are extremely soft; that is, perhaps, their defining feature. Among the softest and most flexible of all technologies in educational systems are pedagogies (methods of teaching). Though pedagogies are the most critical and defining technologies in any assembly intended to teach, they never come first because they must fit in with harder technologies around them; in an institutional context, this includes regulations, timetables, classrooms or learning management systems, the needs of professional bodies, assessment requirements, and so on [7]. Now that we have machines that can play those soft roles of enacting pedagogies, they must do so in the context of what exists. Inevitably, therefore, they start by fitting into those existing structures rather than replacing them. This is, for example, proving to be problematic for teachers who have not adapted their slower changing assessment processes to allow for the large-scale use of LLMs in writing assignments, although such approaches have long been susceptible to contract cheating, including uses of sites such as CourseHero to farm out the work at a very low cost. It is telling that a large majority of their uses in teaching are also meant to replace soft teaching roles, such as developing course outlines, acting as personal tutors, or writing learning outcomes. The fact that they can do so better than an average teacher (though not yet as well as the best) makes it very alluring to use them, if only as a starting point. The fact that they are able to do this so well, however, speaks to the structural uniformity of so many institutional courses. The softness that GAIs emulate means that it is not quite a cookie-cutter approach, but the results harden and reinforce norms. This is happening at a global scale.

Right now, for all of the widely expressed concerns about the student use of AIs, it is easy to see the benefits of using them to support the learning process, and to integrate them fully into learning activities and outcomes. Indeed, it is essential that we do so, because they are not just reflections of our collective intelligence but, from now on, integral parts of it. They are not just aides to cognition but contributors to it, so they must be part of our learning and its context. There are also solid arguments to be made that they provide educational opportunities to those who would otherwise have none, that they broaden the range of what may be taught in a single institution, that they help with the mundane aspects of being part of a machine so that teachers can focus on the softer relational human side of the process, that they can offer personal tuition at a scale that would otherwise be impossible, and that they therefore augment rather than replace human roles in a system. All of this is true today.

Here at the cusp of the AI revolution, we have grown up with and learned to operate those technologies that LLMs are now replacing, and our skills that they replace remain intact. This situation will change if we let it. In the first place, the more soft roles that the machines take on, the less chance we will have to practice them ourselves, or even to learn them in the first place. It is important to emphasize that these are not skills like being able to sharpen a quill or to operate a slide rule, where humans are enacting hard technologies as part of another orchestration. These are the skills for which we develop such hard techniques: the creative, the situated, and the idiosyncratic techniques through which we perform the orchestration, and that are central to our identities as social beings.

Secondly, simple economics means that, if we carry on using them without making substantial changes to the rest of the educational machine, AIs will almost always be cheaper, faster, more responsive, and (notwithstanding their current tendency to confidently make things up) more reliable. In an endemically resource-hungry system, they will be used more and more and, as long as all we choose to focus on are the explicit learning outcomes, they will most likely do so more effectively than real humans. Discriminative AIs will measure such outcomes with greater speed and consistency than any human could achieve; they already can, in many fields of study.

To make things worse, current LLMs are largely trained on human-created content. As the sources increasingly come from prior LLMs, this will change. At best, the output will become more standardized and more average. At worst, the effect will be like that of photocopies of photocopies, each copy becoming less like the original. Fine-tuning

by humans will limit this, at first, but those humans will themselves increasingly be products of an educational system more or less mediated by AIs. Already, there are serious concerns that the hidden guidelines and policies (which are themselves technologies) of the large organizations that train LLMs impose tacit cultural assumptions and biases that may not reflect those of consumers of their products [6], and that may challenge or systematically suppress beliefs that are fundamental to the identities of large numbers of people [68]. The fact that the ways this happen are inscrutable makes this all the more disturbing, especially when ownership of the systems lies in the hands of (say) partisan governments or corporations. There is much to be said for open LLMs as an antidote to such pernicious consequences.

The changes to our individual and collective cognition that result from this happening at scale will be a hard-to-predict mix of positives and negatives; the average capability to do stuff, for instance, will likely improve, though perhaps the peaks will be lower and maybe valuable skills like political reasoning may be lost [5]. It is fairly certain, however, that such changes will occur. Unless we act now to re-evaluate what we want from our education systems, and how much of our soft cognition we wish to offload onto machines, it may be too late because our collective ability to understand may be diminished and/or delegated to smarter machines with non-human goals.

## 6. Discussion: Reducing the Risks of GAIs in Education

There is a wave of change being wrought by the widespread availability and the increasing ubiquity of GAIs, and it makes little sense to stand still as it breaks. We might channel it in useful directions if we had the time but, for now, the large and slow-moving structural changes that this would entail make it difficult, especially while the wave is breaking. This final section presents a few theoretically informed ways that we might surf the wave, taking advantage of the benefits without diverting it or standing in its way.

### 6.1. Partners, Not Tools

The central concern expressed in this paper is that, because GAIs are capable of closely mimicking soft technique, there are great dangers that they will replace not only the mechanical aspects of cognition but the softer cognitive skills required to use them in both teachers and learners. While, from a task completion perspective, it makes a great deal of sense to delegate tasks we cannot do well ourselves, in a learning context this may strongly discourage learners from ever learning them. Whether this is harmful or not depends on the context. For instance, as someone who has spent countless hours for over six decades trying to develop hard skills of drawing, including with the help of digital drawing tools, the author is resigned to the fact that he will probably never learn to do so sufficiently well or quickly enough for it to be a practical option for him beyond personal sensemaking or quick and dirty communication of ideas with others. It therefore seems reasonable for him to delegate illustrations of (say) slide shows or book figures to a GAI. However, it is a very different matter for a child who may never have attempted to learn such skills in the first place. While there are, at least for now, many skills needed to choose and make effective use of GAI image generation tools, so it is not an uncreative act, there are many ways in which drawing with a physical stylus or pen positively affects cognition that will be lost or diminished if this becomes the primary means of doing so. It is important to emphasize that this is not the same as, say, replacing the ability to draw straight lines with a ruler with a drawing program; the skills in jeopardy are the soft, creative, generative, intangible, constructive skills that are a part of, a creator of, and an expression of our cognition itself. This is not a repetition of the error Socrates relates when, in Plato's Phaedrus [69], he says of writing, "this discovery of yours will create forgetfulness in the learners' souls, because they will not use their memories; they will trust to the external written characters and not remember of themselves." The reality is, of course, that writing provides a scaffold, not a replacement for memory: it is a cognitive prosthesis, not an alternative to cognition. However, because GAIs actually do replace the soft skills, it is no longer so clear-cut. Later

in the same passage (ibid. p. 88) Socrates goes on to say, "writing is unfortunately like painting; for the creations of the painter have the attitude of life, and yet if you ask them a question they preserve a solemn silence." A GAI may not remain so silent.

There is, though, a case to be made for the use of AIs in *supporting* a process of drawing (or writing, or making videos, and so on). Tools such as Stable Doodle, Fotor's AI Sketch, or Picsart's SketchAI can take a sketch and turn it into any number of different genres of art or image style, for instance, adding hard skills that the human creator may not have or may not have time to use. The roles they play are not dissimilar to those of the skilled technical teams supporting architects such as Frank Gehry, whose buildings benefit greatly from computer-assisted (and sometimes computer-generated) designs despite his own inability to operate a computer. He relies upon his sketches and rich dialogues with his team to turn his ideas into workable designs for buildings. The important and generalizable point is that there remains scope for soft, creative technique in the process. Similar tools, such as Grammarly or WordTune, that can perform copy-editing roles on human-written text, can be particularly valuable for those writing in a second language, and may help to scaffold the learning of such skills in the first place, without diminishing the creative, generative, soft technique of the writer. This division of roles suggests fruitful ways that we may gain the benefits of AI without losing the essential human engagement and value of the process. As a general principle it is thus better to treat GAIs as partners rather than tools, or as team members or contract workers rather than devices. This makes it easier to divide the cognitive tasks, maintaining human connection where human connection matters. This applies as much to teachers using AIs to support the development and running of a course as it does to the students' studies—in effect, it is now possible for all work to be teamwork. Ideally, more than one GAI should be a team member to reduce the effects of systematic biases and assumptions any one might hold.

What this implies for humans who, in principle, might have performed those roles in the past remains a matter for concern. For a teacher who would otherwise not have a hope of ever being able to assemble or employ the services of a professional design team, and thus the choice lies between receiving an AI's assistance or doing what they can alone, the case for employing an AI is very compelling. At scale, though, this may not bode well for professionals who do currently play those roles and, without them, there will be nothing new to feed the training of the next generation of AI. We can only hope that future generations will still value—and perhaps increasingly value—the work of verifiable humans, for all the reasons previously discussed, though the inequalities and "analogue divide" that may ensue would make this a double-edged sword.

### 6.2. Designing for Intrinsic Motivation

Our educational institutions have evolved to be structurally antagonistic to intrinsic motivation due to the deeply entangled path dependencies embedded in their origins, which has resulted in the phenomenon that many of our most cherished pedagogies and processes are counter-technologies that aim to restore or replace what is lost [7]. The reasons for this are essentially technological, and driven by dynamics of technological evolution described earlier. For our ancestors wishing to share the knowledge of the few with the many, prior to the widespread availability of books and the skills to read them, lectures were the only practical technology. The structural technologies of education systems were therefore primarily developed to make lectures as effective as possible. Timetables, terms, semesters, courses, classrooms that placed lecturers at the front, rules of behaviour for those classrooms, and a host of other technical solutions to this problem therefore became the basis on which all further development occurred, to the extent that they soon became among the hardest and thus the most structurally determinant technologies in the system. Out of necessity, such technologies reduce autonomy for the learners, who must acquiesce to a time, place, pace, and subject matter of someone else's choosing, doing so in an environment where control of almost every second lies with a figure of authority. Unfortunately, autonomy is what self-determination theory shows to be one of the three

essential foundations for intrinsic motivation, without which it cannot occur at all [70]. Furthermore, the need for learning to occur in lock step with other students in a class means that, without much pedagogical ingenuity and skill on the part of the lecturer, some will be bored and others confused, undermining the second pillar of intrinsic motivation, the need for competence/achievable challenge. Only the third foundation, relatedness, is potentially well supported, if only thanks to the presence of other learners in the same situation. As a result, many of our most cherished pedagogies, from problem- or inquiry-based learning to direct instruction and the chunking of content, are focused on ways of restoring autonomy and supporting individuals' development of competence. This demands a lot of work, hard skill, and soft talent from a sensitive and hard-working teacher or (at greater cost) teachers, albeit that the work is assisted by campus designs that make social interaction almost unavoidable.

A more reliable, superficially cheaper, and less demanding way of ensuring students do the necessary work to learn from designated teachers is through the use of extrinsic motivation such as grades, the promise of credentials, rules of attendance, and so on, and these forms of coercion have therefore also become hard structural elements of most educational systems. Unfortunately, extrinsic motivation invariably crowds out and, at best, permanently diminishes intrinsic motivation [60,70,71], making the reward or avoidance of punishment the primary purpose of learning. To make matters worse, it could not send a stronger message that an activity is undesirable if a reward is given for its accomplishment, or punishment for failure to accomplish it [60]. One major consequence of this is that an intelligent student, whose intrinsic motivation has been diminished by the reward or punishment, and who has been given every indication that achievement of the grade is the primary purpose for attending, will take the shortest path to achieve it. This in turn leads to cheating, satisficing, and limited risk taking (ibid). It is not surprising that students use generative AIs to assist with or perform such tasks. Simply developing counter-technologies to this is an endless arms race that no one can win [72,73], and all such technologies, from proctored exams to learning diaries or other products that reveal the process, can only ever be temporary solutions that hold until further counter-technologies are available to defeat them. An LLM can easily be persuaded to provide convincing personal reflections or work in progress. Many technologies are available to connect with them in proctored exams, and these will only improve. For every technology we create to prevent cheating, as long as the purpose is perceived as achievement of grades or credentials, counter-technologies will be invented to overcome it.

While ungrading approaches [74] that focus on feedback rather than extrinsic drivers can reduce the harm, as long as credentials remain structurally embedded as the primary purpose of learning, the problem will persist. To break this cycle, any effective structural solution should therefore start with decoupling learning and credentials. There are many ways that this may currently be achieved, even within existing educational models. The Biomedical Sciences program at Brunel University, for example, divides programs into study blocks (courses), which are ungraded, and integrative assessment blocks, that integrate knowledge and skills from across the study blocks and that provide evidence for which qualifications are awarded [75]. Athabasca University provides challenge assessments for courses that permit students to study independently and/or use their existing knowledge that may be used in a similar way. Even within a conventional course, grades may be avoided until absolutely necessary. While the potential for taking shortcuts remains almost as great in those assessments as in courses with tightly coupled learning and assessment, the study process itself remains largely free of such concerns, notwithstanding risks of teaching to the test, and thus it becomes possible to design structures and supports in ways that better support intrinsic motivation, that support risk taking, that allow failure to be intrinsic to the process, that valorize diversity, and that do not need to be so tightly bound to measurable outcomes. Meanwhile, once credentials are decoupled from the learning process, greater focus may be given to making assessments more personally relevant, reliable, authentic, and effective, especially when, as in Brunel's model, the assessments are

challenging, useful, authentic, integrative learning experiences in their own right. Other autonomy-restoring pedagogies may help even if structural changes are difficult to make, such as allowing students to have a say in the development of learning outcomes, giving them agency in the assessment itself, or simply designing a process that allows students autonomy in the selection of methods, outputs, or media. Softness for students is a prerequisite for autonomy, notwithstanding that some constraints and boundaries are essential for creativity [76]. The issue is not whether educational systems should be hard but which parts we choose to harden.

Social interaction is also critical to supporting intrinsic motivation, as well as building relationships, modelling ways of being, and addressing many of the broader, softer social goals of education. When teachers (including other students) are engaged with students throughout the process, learning itself (rather than its terminal products) becomes visible, cheating is far less likely and more difficult to accomplish, and students can exercise more control over their learning journeys. If AIs are involved in this, rather than replacing didactic teaching roles they can be trained to facilitate such interactions, prompting connections, weaving threads of conversation together, encouraging dialogue, summarizing discussions, arranging meetings, and so on [77].

Finally, for all of the risks, there is a role for AIs in supporting needs for competence through the development of hard technique, acting as patient, knowledgeable partners able to explain things in ways that a student will more easily understand, offering feedback, providing challenges appropriate to needs, filling in background knowledge, prompting conversations, developing personalized activities, and even (in limited ways) modelling effective practices. Careful prompting and fine-tuning may be needed to avoid risks of encroaching too far into the softer territory that is or should remain the domain of humans, and opportunities should always be sought to ground what is learned in a human, social context. Exactly what those hard techniques they teach might be will vary according to subject needs, and they may include those that GAIs can better do themselves. For example, when we ask an LLM to write code for us, it may help us better understand how to code ourselves, but it raises the question as to why we would bother in the first place. This is a challenge. As our lives increasingly integrate GAIs there will be some skills that are habitually delegated to them, so it may make little sense for those who argue that education should be seen in terms of hard, measurable outcomes to learn them or teach them. Their arguments will be compelling; whether or not we have concerns about the human abilities they therefore replace, education is a preparation for life and, if machines are ubiquitously parts of our lives, it would be Quixotic to insist on learning skills that will never be used. However, it is important to remember the hidden purposes and tacit utility that bring softness to even the hardest of technologies, and the many ways that technologies can be assembled to perform tasks far beyond the intents of their designers. There is intrinsic value to be found in overcoming challenges and developing competence, even when it is something as simple as sawing wood, washing dishes, or playing musical scales with precision, and even when it is something machines could do more cheaply, more effectively, and faster. Each time we lose or fail to learn a hard skill, it shuts down the unprestatable, unpredictable adjacent possibles that it might have provided. Again, this speaks to the central point of this paper: the purpose of education is not the acquisition of skills and knowledge. Those are just some of the means through which it is accomplished. The purpose of education is the development of human beings and the societies they live in.

## 7. Conclusions

It makes no more sense to avoid using AIs in both teaching and learning than it does to avoid using words. These technologies already are a part of the fabric of our shared, technologically mediated cognition and, whether we like it or not (barring catastrophic disasters), they can and will play substantial roles in what and how learning happens, both formally and informally, in all walks of life. The question is not whether but how they will play those roles. Quite apart from pragmatic and ethical concerns about how they

are trained, who owns them, and how they can become less unreliable, AIs pose many potential threats to all aspects of our social, political, and personal lives, from the loss of jobs to the collapse of economies to the end of the human race [78], and much thought is needed to find ways those risks can be mitigated or forestalled. However, though less dramatic, less immediate, and more insidious, the effects on the things that make us who we are—our intelligence, our creativity, our relatedness, our identities—are perhaps the most dangerous of all. If, as they must, AIs affect how and what we learn, it will change us as a species, in ways that (as this paper has shown) may be far from beneficial.

In the first place, there is an urgent need for more research that focuses on the tacit, implicit, and systemic effects of education rather than its effectiveness in achieving intended learning outcomes, so that we are better able to identify changes as they occur. Although standardized approaches to measuring creativity and intelligence may provide some indicators of change, the results of such measures fluctuate for many reasons apart from educational experiences, so they will tell us little. By and large, the effects will be hard to quantify and impossible to replicate because of the situated, complex nature of the process. Soft research approaches such as outcome harvesting [79], appreciative inquiry [80], storytelling, soft systems methods [81], rich case studies, and grounded theory may help to reveal some of the effects of the hidden curriculum, and to help establish baselines against which future learners may be compared.

More broadly, now would be an excellent time to do as many have advocated for over 100 years and re-evaluate the purpose, form, and function of our educational systems. However, educational institutions are deeply entangled with many aspects of societies, any or all of which are mutually hardening, making them highly resilient to major change, at least in less than a generation or two. Although it would be desirable to redesign our institutions from scratch, we cannot simply and unilaterally abandon structural motifs like courses, credentials, timetables, curricula, systems of credit transfer, exams, programs, or rules of attendance, not to mention all of the supporting infrastructure, without instigating an economic and social disaster of unimaginable proportions.

One choice available to us—the easy choice—is to think locally, to solve problems as they emerge in piecemeal fashion, and to develop counter-technologies to address the disruption; legislation, AI-detection tools, increasingly stringent proctoring processes, and so on may indeed put a Band-Aid over holes that appear before they get too large. However, this is, as Dubos [82] put it, a philosophy of despair, as each counter-technology spawns its own counter-technologies in an endless spiral. It would be better to think structurally and globally about ways of embracing rather than resisting the change.

Our best option for now seems to be to find ways to work with AIs as partners, team-mates, and contractors, and to focus on uses that augment rather than replace the things that we most value educationally, personally, and socially; uses where their capacity for soft technique complements but does not replace our own. It would be very helpful if governments and other sources of funding and accreditation that play some of the hardest structuring roles, and that often seem intent on treating institutions solely as economic drivers and creators of productive workforces, focused more clearly on the more fundamental value of education as both a stabilizing and a creative force in society, being one that supports cultural as well as economic goals, and one that makes life better, richer, more rewarding, and safer for everyone. However, even if that does not occur, we can still structure what we already have so that the extrinsic drivers that shape attitudes, processes, and beliefs about education lose some or all of their power. We can seek ways of using our new, tireless partners to connect us, to empower us, and to support rather than control us. We can study, acknowledge, and integrate the changes that AIs bring across the workplace and society, and we can search for, examine, debate, and nurture the sacrosanct spaces, the things that we cannot or should not (at least yet) let go. Radical change wrought by the growth in reach and power of AI is now all but certain, so there is some urgency to this. It is the job of this generation, living at a transition point in the history of the human race, to

create structures that preserve what must be preserved, as much as it is to embrace what must be changed.

**Funding:** This research received no external funding.

**Data Availability Statement:** Not applicable.

**Conflicts of Interest:** The author declares no conflict of interest.

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
