# Peer review of "The Human Nature of Generative AIs and the Technological Nature of Humanity: Implications for Education"

_digital, doi:10.3390/digital3040020_

Round 1

Reviewer 1 Report

Comments and Suggestions for Authors

The purpose of the study described in the passage is to examine and analyze how the widespread use of Generative AIs (GAIs), particularly in the context of education, will impact and transform various aspects of human behavior, cognition, and collective intelligence. The study aims to explore the profound consequences of GAIs in education and collective intelligence and suggests theoretically informed approaches that may help mitigate potential risks while harnessing the benefits of generative AIs. In essence, it seeks to understand the transformative impact of GAIs on human society, particularly in the realm of education and collective intelligence, and offer insights into how we can adapt and navigate these changes effectively.

In summary, the study not only addresses existing gaps and clarifies concepts. While the study does not explicitly outline specific gaps in existing literature, it hints at several research areas by addressing the transformative nature of GAIs, the integration of soft techniques, the interconnectedness of education and collective intelligence, the use of a theoretical framework, etc.

The consistency of the conclusions with the arguments presented in the research study is a crucial aspect of research validity. The conclusions logically follow from the analysis presented throughout the study. Additionally, the conclusions are directly address the main research issues posed at the beginning of the study.

The fact that the study utilizes 62 references, most of which are recent, is indicative of a thorough and up-to-date literature review. This demonstrates that the research is well-informed and has considered the latest developments and contributions in the subject area. A substantial number of recent references also indicate that the study is building upon current research trends and addressing contemporary issues in the field.

Author Response

Thanks for the very positive review! I don't think any changes are required.

Reviewer 2 Report

Comments and Suggestions for Authors

The paper takes on an important and highly relevant topic that of the widespread use of generative AI systems like ChatGPT in education. It argues that generative AIs represent a completely new phenomenon in human history because unlike previous technologies, they can mimic the "soft techniques" of human cognition like creativity and problem-solving. This means AIs can now perform roles that were previously the sole domain of people. It strikes a cautionary note in that an over-reliance on AI risks diminishing development of human skills – and raises fundamental questions about what it means to be human. The author suggests that working with AI may prevent students from gaining important tacit skills and knowledge.

That said, the paper could have benefited from a more nuanced discussion of the differences between AI and human cognition. There is an intentionality to human cognition as well as a connection to the real world that is missing in the case of AI. For instance, though AI can spout verbiage about any real-world concept it truly does not understand what it is referring to. Moreover, the fundamental claim that CI will change human learning and cognition, is not necessarily substantiated by any strong argument. There are many different ways that AI will play out in education and outside of it and some exploration of those alternative possibilities would have been helpful to make the argument a bit more complex.

If there is one fundamental flaw in the paper it is that it give a somewhat simplistic “cause-effect” relationship between the powers of AI and its impact on the world. What AI is (or can be or will be) will depend on a wide range of socio-political-economic design decisions, some of which will be made by the companies and some will be made in ad-hoc, historically contingent ways. There is nothing inherent in AI that leads to one specific end result. For instance, one could have imagined a very different future for social media – but what it is and has become is the result of being embedded in a particular economic system that values time on site and engagement over broader social goals.

Finally, a criticism of the style of writing of the paper. It is written in a highly abstract, jargon-laden style that will render the ideas inaccessible to many readers. This is an important paper for a range of reasons, not the least of which are the real-world implications of AI in education. Many of the ideas here could have greater impact if it were written in a simpler, more accessible style.

Comments on the Quality of English Language

See last paragraph of review.

Author Response

Thanks for the helpful review and thoughtful comments.

I agree about the large differences between AI and human cognition, and already tried to discuss the more important ones, including issues of intentionality and meaning (see the section starting at line 209 in the original draft), but I have tried to add nuance to the discussion by drawing more attention to possible similarities and why those similarities are insufficient to treat them in the same manner. It is a complex issue that really demands its own paper and there are arguments on both sides so I have not exhausted the possibilities, dwelling mainly on the most significant point that, no matter what the other similarities might be, an LLM uses nothing but second-hand signals created by humans so cannot be treated as in any meaningful way similar to a human - it's a reflection of our own collective intelligence, not an independent thinking thing with an identity and purposes of its own. As the purpose of this section is just to establish that there are differences, I hope that is enough. If there are such differences, then the fears expressed about learning to be human from machines stand.

I have also slightly bolstered the discussion of the many different ways that AIs may entangle with our lives beyond those mentioned, though it is difficult to know where to stop - again, there's another whole paper in that. I take the point about the complexity of the socio-technical systems in which AIs are entangled and have often made such points myself - indeed, this is one of the cornerstones of the book from which the theory in this paper is derived. However, the point of drawing attention to why genAIs are fundamentally different is that that there are some generalizations that can be made as a result. Although I had already mentioned this, I have made a stronger point in the new methodology section that we are in the realm of massively complex systems and, following Kauffman, that the consequences are inherently unprestatable, so any predictions I make should be taken with a pinch of salt. I have included some further mention of a few of the creative possibilities to help make the view more balanced.

  I have attempted to make the language a bit more accessible, here and there and where possible, including explaining a few of the terms and adding a couple of subheadings to help navigate the arguments, though it is difficult to entirely avoid jargon in an area like AI! There are also one or two places that I have simplified my own use of English.

Reviewer 3 Report

Comments and Suggestions for Authors

The article presents the following general concerns:

  • Despite being a review article, the abstract must mention the main findings and results of the research, in addition to the methodologies implemented to carry it out.

  • The contribution of this research needs to be clarified. Please clarify.

  • Please avoid using the pronoun I. Instead, use the passive voice.

  • The introduction is relatively short, extended to one page, and address: General context: Present the general scope of your review, offering a context so that the reader understands the topic. Topic Importance: Why is it relevant or meaningful to review this particular topic? Is there a specific need in the field or a gap in existing knowledge that requires a review? Objective: Clarify what the primary purpose of your review is. Selection criteria: This may include years of publication, specific methodologies, geographic location of the studies, etc. Limitations and delimitations: Clarify the scope of your review. Are there aspects of the topic that you will need to address? If so, it is helpful for the reader to know that from the beginning. Review Structure: Provide a brief outline of how the rest of your review is structured. This helps the reader anticipate how your analysis and argument will develop. Key definitions: If specialized terminology or key concepts are central to your review, now is an excellent time to define them and contextualize them for your audience. Current state of knowledge: Depending on the length of your introduction and the type of review, you could offer a summary of the current state of knowledge or debate around the topic you are reviewing.

  • Add a methodology section, which expands, clarifies, and details how the search for the articles was carried out, what the acceptance and rejection criteria were, and how the analysis was carried out.

  • Sections 2 and 3 require better referencing.

  • The presented article has several significant areas for improvement that compromise its quality and relevance. Firstly, the abstract does not provide information on the main findings, results, or methodologies used. Furthermore, the unique contribution or relevance of the article to its field of study must be clearly defined, leaving the reader in doubt about its actual contribution. Likewise, the introduction is highly brief and omits crucial aspects, such as the general context, the relevance of the topic, and the purpose of the review, among other vital elements that establish the foundation of the study. A particularly worrying point is the need for a methodology section. Without this section, evaluating the rigor of the research process and the study's replicability is difficult. Finally, a discussion section is necessary to ensure proper interpretation and contextualization of the results. Therefore, they justify its rejection in its current form.

  •  
  • From my point of view the introduction must contain, before line 29, certain references about the usage of AI in different areas of the knowledge, then, use the following references to justify this fact: Tendency on the Application of Drill-Down Analysis in Scientific Studies: A Systematic Review; Implementation of ANN-Based Auto-Adjustable for a Pneumatic Servo System Embedded on FPGA; Neural Network and Spatial Model to Estimate Sustainable Transport Demand in an Extensive Metropolitan Area; A Deep Learning Approach for Predicting Multiple Sclerosis; A Novel Methodology for Classifying Electrical Disturbances Using Deep Neural Networks; Spatial Models and Neural Network for Identifying Sustainable Transportation Projects with Study Case in Querétaro, an Intermediate Mexican City

  • Add a discussion section.

Comments on the Quality of English Language

The following misspellings should be checked:

  1. line 26: “In this paper I…” It seems that you are missing a comma. Adding a comma after paper.

  2. line 65: The use of “and/or” is severely frowned upon in formal writing. Consider using only one conjunction or rewriting the sentence. 

  3. line 167: The abbreviation “e.g” seems to be incorrectly punctuated. Consider changing the punctuation by “e.g.”

  4. line 203: The phrase “are capable of being” may be wordy. Consider changing the wording by “ca be”

  5. line 214: Some readers may consider this use of “blindly” insensitive. Consider changing. 

  6. line 229: “It could be argued that none of this is important…” Hedging language such as “I think…” and “it would be great” can come across as uncertain or indecisive. Clear, deliberate phrasing conveys confidence and may improve how your writing is received. 

  7. line 236: “written by a Gustav Mahler…” It seems that there is an article usage problem here. Eliminate the article. 

Author Response

Thanks for the thoughtful and detailed review. Responses inline:

 Despite being a review article, the abstract must mention the main findings and results of the research, in addition to the methodologies implemented to carry it out.

Done

The contribution of this research needs to be clarified. Please clarify.

Done.

Please avoid using the pronoun I. Instead, use the passive voice.

Done (with some reluctance - my own role in developing the theory makes use of the personal pronoun appropriate).

The introduction is relatively short, extended to one page, and address: General context: Present the general scope of your review, offering a context so that the reader understands the topic.

Done. The introduction is longer and now situates the paper in the context of the problem and other discussions of it.

Topic Importance: Why is it relevant or meaningful to review this particular topic?

Done, as above.

Is there a specific need in the field or a gap in existing knowledge that requires a review?

Done at the start of the introduction.

Objective: Clarify what the primary purpose of your review is.

Done, as above.

Selection criteria: This may include years of publication, specific methodologies, geographic location of the studies, etc.

This is not a literature review but an argument that draws on literature from fields as diverse as machine learning, neuroscience, the philosophy of technology and socio-technical systems, complex systems theory, and education. As such the selection criteria are hard to pin down. I have situated the argument in its field.

Limitations and delimitations: Clarify the scope of your review. Are there aspects of the topic that you will need to address? If so, it is helpful for the reader to know that from the beginning.

Done - the methodology now makes it clearer that the premises of the argument itself imply that its predictions will most likely be wrong, and that the intent is to provide a means of understanding the future, not predicting it.

Review Structure: Provide a brief outline of how the rest of your review is structured. This helps the reader anticipate how your analysis and argument will develop.

I've tried to make this clearer in the extended introduction.

Key definitions: If specialized terminology or key concepts are central to your review, now is an excellent time to define them and contextualize them for your audience.

Not really relevant here, I think - at least, the ones that really matter have to be explained in a narrative because of dependencies on prior concepts.

Current state of knowledge: Depending on the length of your introduction and the type of review, you could offer a summary of the current state of knowledge or debate around the topic you are reviewing.

This is helpful. I have summarized this in the introduction, when explaining why my own analysis is worth doing.

Add a methodology section, which expands, clarifies, and details how the search for the articles was carried out, what the acceptance and rejection criteria were, and how the analysis was carried out.

This makes no sense in this context because it is not a literature review.

Sections 2 and 3 require better referencing.

Section 2 (now 3) is a summary of the theory in my book, in the most concise form I could manage. It has references (quite a few) where appropriate and any claim not referenced (as far as I know) comes from the book itself. Rather than repeatedly reference the book I have explained this at the start of the section.

I have tried to find unsubstantiated claims in section 3 (now 4) and added references to support them.

The presented article has several significant areas for improvement that compromise its quality and relevance. Firstly, the abstract does not provide information on the main findings, results, or methodologies used.

now added.

Furthermore, the unique contribution or relevance of the article to its field of study must be clearly defined, leaving the reader in doubt about its actual contribution.

now added.

Likewise, the introduction is highly brief and omits crucial aspects, such as the general context, the relevance of the topic, and the purpose of the review, among other vital elements that establish the foundation of the study. A particularly worrying point is the need for a methodology section. Without this section, evaluating the rigor of the research process and the study's replicability is difficult.

This is a grounded argument, not a replicable study or literature review. Hopefully this is now clearer.  If the reader follows the argument then they can take it on its own merits. I have added the methodology section as requested.

Finally, a discussion section is necessary to ensure proper interpretation and contextualization of the results.

The whole paper is a discussion section. I have, though, labelled the section discussing measures that can be taken to limit potential risks as a discussion section.

Therefore, they justify its rejection in its current form. From my point of view the introduction must contain, before line 29, certain references about the usage of AI in different areas of the knowledge, then, use the following references to justify this fact: Tendency on the Application of Drill-Down Analysis in Scientific Studies: A Systematic Review; Implementation of ANN-Based Auto-Adjustable for a Pneumatic Servo System Embedded on FPGA; Neural Network and Spatial Model to Estimate Sustainable Transport Demand in an Extensive Metropolitan Area; A Deep Learning Approach for Predicting Multiple Sclerosis; A Novel Methodology for Classifying Electrical Disturbances Using Deep Neural Networks; Spatial Models and Neural Network for Identifying Sustainable Transportation Projects with Study Case in Querétaro, an Intermediate Mexican City

I have looked up these references and notice that they all appear to come from the same team at Universidad Autónoma de Querétaro (and often from the same people). Some of the papers are interesting. However, I am not sure why these were suggested because I can see no way that they support or challenge any of the arguments presented, they have nothing to do with the field of education, and they are at best tangential to the issues discussed here. Perhaps I will use them in another paper.

Add a discussion section. ;

Done (see above).

Detailed typos, grammatical, and spelling concerns:

1) Rephrased.

2) I don't think it was ambiguous but I have changed it to a wordier form.

3) done.

4) done.

5) changed to "unknowingly"

6) changed to "some might argue"

7) done.

Thanks again for this. It has been challenging but rewarding  to answer these concerns!

Reviewer 4 Report

Comments and Suggestions for Authors

Dear Authors. 

I find the research you propose very interesting. 

The introductory section needs to be reformulated. Add a greater number of citations to better contextualize it. 

Add practical applications of this study. 

Add limitations derived from this research. 

Revise the wording of the article. Avoid using the first person singular. 

Author Response

I have extended the introduction to provide more context, with some references, and to explain the purpose more clearly. I think it reads better now.

I have added a methodology section discussing limitations.

I have removed first person singular references.

I have tried to make the recommendation section (now labelled as a discussion) more clearly indicate the practical applications of this way of understanding AI, as well as some more discussion of the potential positive benefits of treating GAIs as partners. However, I found it difficult to do much more than I had already done because the discussion section is really about nothing else than practical applications of the study.

Round 2

Reviewer 3 Report

Comments and Suggestions for Authors

The manuscript can be published.

Author Response

Thank you!